# The Incidence of Intestinal Gastric Cancer among Resettlers in Germany—Do Resettlers Remain at an Elevated Risk in Comparison to the General Population?

**DOI:** 10.3390/ijerph17249215

**Published:** 2020-12-09

**Authors:** Anna Lindblad, Simone Kaucher, Philipp Jaehn, Hiltraud Kajüter, Bernd Holleczek, Lauren Lissner, Heiko Becher, Volker Winkler

**Affiliations:** 1Heidelberg Institute of Global Health, University Hospital Heidelberg, 69120 Heidelberg, Germany; anna2.lindblad@gmail.com (A.L.); simone.kaucher@uni-heidelberg.de (S.K.); 2Institute of Social Medicine and Epidemiology, Brandenburg Medical School Theodor Fontane, 14770 Brandenburg an der Havel, Germany; Philipp.Jaehn@mhb-fontane.de; 3Cancer Registry, North Rhine-Westphalia, 44801 Bochum, Germany; Hiltraud.Kajueter@krebsregister.nrw.de; 4Saarland Cancer Registry, 66119 Saarbrücken, Germany; b.holleczek@krebsregister.saarland.de; 5School of Public Health and Community Medicine, Institute of Medicine, Sahlgrenska Academy at University of Gothenburg, 41346 Gothenburg, Sweden; lauren.lissner@gu.se; 6Institute for Medical Biometry and Epidemiology, University Medical Center Hamburg-Eppendorf, 20246 Hamburg, Germany; h.becher@uke.de

**Keywords:** incidence, stomach cancer, Laurén classification, migrants, former Soviet Union, cohort, Germany

## Abstract

Objective: Previous studies have shown that the incidence of gastric cancer (GC), and particularly intestinal GC, is higher among resettlers from the former Soviet Union (FSU) than in the general German population. Our aim was to investigate if the higher risk remains over time. Methods: GC cases between 1994 and 2013, in a cohort of 32,972 resettlers, were identified by the respective federal cancer registry. Age-standardized rates (ASRs) and standardized incidence ratios (SIRs) were analyzed in comparison to the general population for GC subtypes according to the Laurén classification. Additionally, the cohort was pooled with data from a second resettler cohort from Saarland to investigate time trends using negative binomial regression. Results: The incidence of intestinal GC was elevated among resettlers in comparison to the general population (SIR (men) 1.64, 95% CI: 1.09–2.37; SIR (women) 1.91, 95% CI: 1.15–2.98). The analysis with the pooled data confirmed an elevated SIR, which was stable over time. Conclusion: Resettlers’ higher risk of developing intestinal GC does not attenuate towards the incidence in the general German population. Dietary and lifestyle patterns might amplify the risk of GC, and we believe that further investigation of risk behaviors is needed to better understand the development of disease pattern among migrants.

## 1. Introduction

Migration is a growing phenomenon in the world. In 2019, 272 million people were international migrants according to the International Organization for Migration [1]. Given diverse push and pull factors behind migration, migrants are a heterogenous group of people [2]. Traditionally, the health of migrants is investigated by comparing it to the health of the host country’s non-migrant population. Upon their arrival, migrants may suffer from less morbidity and mortality as a result of the “healthy migrant effect”. In general, differences in health and disease are expected to attenuate over time as migrants adapt to exposure of risk factors and changing environments [3,4]. As exposures change over time, the importance of a life course approach has been emphasized for further understanding migrant health and its disparities [5].

In 2018, one out of four people living in Germany had a migration background [6]. To date, the second largest migrant group consists of ethnic German resettlers whose ancestors migrated to Russia in 18th and 19th centuries on invitation by the Russian empress to farm unsettled land. After the Second World War, ethnic Germans and their families were invited by the government of West Germany to return to Germany. They obtained German citizenship upon arrival and were quasi-randomly allocated to the federal states based on population density and economic conditions. Due to strict emigration regulations, it was not until after the collapse of the Soviet Union that a significant migration flow reached Germany peaking in 1994 with more than 200,000 people per year. Until 2019, about 2.4 million resettlers have migrated from the former Soviet Union to Germany [7,8,9].

Gastric cancer (GC) is the fifth most common cancer type worldwide, and it occurs twice as frequent among men than women [10]. The prognosis is poor: in Germany the relative five-year survival rate is 30–35% [11]. GC is a heterogeneous disorder, associated with both genetic and environmental factors. The most important risk factor is infection with *Helicobacter pylori* (*H. pylori*). Lifestyle and dietary risk factors associated to GC are alcohol, smoking, red and processed meat, salty foods, obesity, and low physical activity. Age, male gender, low socioeconomic status, and gastroesophageal reflux are other factors predisposing GC [12,13,14,15,16]. The incidence of GC has decreased significantly over the last decades, probably explained by higher standards of hygiene, better nutrition, and eradication of *H. pylori* infections [17,18]. According to the widely used Laurén classification, GC can be distinguished into two main histologic types: diffuse and intestinal. Briefly, the intestinal type is composed of well-differentiated polarized cells forming glandular structures, whereas the diffuse type consists of less differentiated and unpolarized cells in randomly ordered cell clusters [19]. Regarding epidemiology, intestinal GC occur predominantly in men and in older age groups, whereas diffuse GC is equally frequent in both sexes and is more common at younger ages [20]. Evidence suggests that the two subtypes are associated to risk factors such as lifestyle and dietary factors to varying degrees [21,22,23].

Previous studies have found an elevated incidence and mortality of GC among resettlers, and particularly a higher risk of developing intestinal GC, in comparison to the general population [24,25,26,27]. The aim of this study was to further investigate differences in intestinal GC among resettlers by (i) replicating previous findings in another cohort of resettlers and (ii) pooling all available data on resettlers’ GC incidence for a joint analysis to provide insight into whether subtype specific risk differences attenuate or remain over time.

## 2. Methods

### 2.1. Münster Cohort

The study is mainly based on a registry-based cohort established in the administrative district of Münster (the AMIN cohort; Aussiedler in Münster—Incidence cohort study). Details about the Münster cohort and the follow up procedures can be found elsewhere [25,28]. In brief, the cohort comprised 32,962 resettlers who were assigned to the district between 1990 and 2001 and is a quasi-random sample of 53% of all resettlers in that district which is a part of the federal state North Rhine-Westphalia (NRW), Germany. Person-years (PY) of time under risk for developing a cancer condition were estimated from the beginning of 1994 or immigration (if later than 1994) to the diagnosis of GC or the end of follow-up (31st December 2013) taking into account deaths and out-migration of the study area [25,29]. The accumulated person-time of the general population of the Münster region was derived from the midyear population (from 1994 to 2013), provided by the federal cancer registry of NRW. The study was conducted in accordance with the Declaration of Helsinki, and the protocol was approved by the Ethics Committee of the Medical Faculty, University Hospital Heidelberg (S-319/2013).

Records of patients with incident GC cases diagnosed between 1994 and 2013 were provided by the NRW cancer registry (three-digit ICD-10 code: C16, ICD-9 code: 151). Cases among resettlers were identified by a pseudonymized two-step record linkage procedure [7,30]. GC cases were divided into three subgroups based on the tumor morphology according to ICD-O-3 (International Classification System of Diseases for Oncology, 3rd edition) and the Laurén classification: intestinal GC, diffuse GC, and other/missing GC. The latter group comprises carcinomas with other specified histology patterns or unspecified or missing tumor morphology. The classification and numbers of cases are summarized in Table 1 (morphology codes according to the second edition of the ICD-O used for cases diagnosed between 1992 and 2002 could be unambiguously translated into ICD-O-3 values) [31].

#### Statistical Analyses

Age-standardized incidence rates (ASR) for the Münster population were calculated by direct standardization using the old European standard population [32]. Calculations were performed by sex and two-year calendar periods (1: 1994–1995, 2: 1996–1997, …, 10: 2012–2013) for the total group and for each histologic GC subgroup separately. Additionally, negative binomial regression was used to model ASR time trends by estimating age-specific rates, which were again standardized directly. The model used the number of cases as the dependent variable and sex (categorical), age group (categorical), and calendar year (continuous, calendar year—1990) as independent variables and log(PY) as the offset. Standardized incidence ratios (SIR) of resettlers in comparison to the general Münster population were calculated using indirect standardization. SIRs were calculated separated by sex and by the three subgroups of GC.

### 2.2. Pooled Data from Münster and Saarland Cohorts

To further investigate SIR and the secular trends in GC incidence among resettlers in comparison to the general population, data from the Münster cohort and the AMOR cohort, a second resettler cohort from the federal state of Saarland, were combined [24]. The characteristics of the pooled data are represented in Table 2, further description of the Saarland cohort can be found elsewhere [26,28]. In brief, the AMOR cohort comprises 18,619 resettlers who were assigned to the federal state of the Saarland, Germany, between 1990 and 2005. Incidence data from 1990 to 2009 were provided by the Saarland Cancer Registry, and vital status follow-up was performed by local registry offices. Data from the mid-year Saarland population were provided by the federal statistics offices. The combined observation time for the Münster cohort and the Saarland cohort was between 1990 and 2013.

#### Statistical Analyses

The expected cases for the pooled SIR analysis were calculated for each cohort separately using incidence and population data of the respective host populations. SIRs were calculated for the whole study time and for the calendar periods 1990–2001 and 2002–2013. Additionally, time trends of SIRs were modeled using negative binomial regression. The model used the number of observed events among resettlers as the dependent variable and sex (categorical), subtype (categorical), calendar year (continuous, calendar year—1990), and cohort (categorical) as independent variables and the number of expected cases as the offset. ASRs and SIRs were calculated with exact 95% confidence intervals (95% CIs) and the significance level was set to 0.05. Statistical analyses were performed using Stata IC (version 14) (StataCorp LLC, Texas, USA).

## 3. Results

### 3.1. Descriptive Results

Table 2 summarizes characteristics of the study population in Münster and the pooled data. The Münster cohort comprised 16,033 men and 16,939 women, and a total of 462,823 PY with a mean follow-up time for 13.4 years. The mid-year population in Münster was 2,558,285 in 1994 and 2,574,148 in 2013. During the study period, 10,873 gastric cancer cases were registered in the Münster region. Intestinal GC represented the most common subtype. For both sexes combined, the median age at diagnosis was 73 years for patients with intestinal GC and 69 years for patients with diffuse GC. The median age at diagnosis was 75 years for patients with other/missing GC. For resettlers, the median age at diagnosis was significantly lower than their counterparts in Münster for diffuse GC (*p* = 0.015) and other/missing GC (*p* = 0.021), but not for intestinal GC (*p* = 0.222). No associations were seen between resettler status and histologic type. The tumor was histologically confirmed in 79% of all cancer cases, and 10% of the cancer cases were notified by death certificate only (DCO).

The pooled cohort data accumulated 667,190 person-years (52% female years). The midyear population was on average 3,055,145. Of 152 reported GC diagnoses among resettlers, 52% were of intestinal type. Median age at diagnosis for both male and female resettlers was also significantly lower than their counterparts in the general population (intestinal GC *p* = 0.014, diffuse GC *p* < 0.001, other/missing GC *p* = 0.003), and again there was no association between resettler status and histologic type (*p* = 0.124 for men and *p* = 0.176 for women).

### 3.2. Münster Population

Figure 1 illustrates ASRs for GC subtypes in the general population of Münster. The ASR for total GC decreased from 24.8 (95% CI: 22.7–26.8) to 16.4 (95% CI: 15.1–17.8) for men and from 13.1 (95% CI: 12.0–14.3) to 8.9 (95% CI: 7.9–9.8) for women from 1994–1995 to 2012–2013. Mean ASR for intestinal GC was more than twice as high among men than women (9.9 (95% CI: 9.5–10.2) and 3.6 (95% CI: 3.4–3.8), respectively) and did not change over time. There was strong evidence for a declining trend for other/missing GC for both sexes (*p* < 0.001). The group of missing/other GC consisted of 73% missing GC cases in 1994 and 54% in 2013.

### 3.3. Resettlers in Münster

Compared to the general Münster population, the SIR of total GC was elevated for male but not for female resettlers (Table 3). Intestinal GC was elevated among both sexes (men: 1.64 (95% CI: 1.09–2.37); women: 1.91 (95% CI: 1.15–2.98)). SIR of diffuse GC varied and was 1.61 (95% CI: 0.88–2.70) for men and 0.64 (95% CI: 0.23–1.39) for women.

### 3.4. Pooled Cohorts Analyses

Between 1990 and 2013, the SIRs of total GC and intestinal GC were elevated for both women and men in the Saarland cohort and Münster cohort combined (Table 4). When dividing the observation time into two periods (1990–2001 and 2002–2013), the incidence was elevated among resettlers in both periods. No difference in incidence was seen for diffuse GC. Regarding other/missing GC, SIR was elevated for female resettlers in the first period, with a major decrease in the second period. In Appendix A, SIRs are presented for each period and cohort separately. The incidence of intestinal GC was elevated in the second period for both sexes in both cohorts. Regarding diffuse GC, the incidence was only elevated among women in Saarland cohort during the second period.

No time trend could be observed (calendar year coefficient = 0.002, *p* = 0.889) when modeling SIR (Table 5). Likewise, no difference in SIR for sex could be seen (female coefficient = 0.025, *p* = 0.886). The SIR for diffuse GC was lower than for intestinal GC (diffuse type coefficient = −0.639, *p* = 0.013), and one could see an effect for cohort (Saarland coefficient 0.558, *p* < 0.001). The estimated SIR for a covariable combination can be obtained; for example, for the year 2000, intestinal GC, female, Münster cohort, the estimate is exp(0.535 + 0.002 × (2000 − 1990) + 0.025) = 1.79.

## 4. Discussion and Conclusions

### 4.1. Key Findings

Our results are largely consistent with previous findings in the AMOR cohort from the federal state of Saarland: we found that the incidence of intestinal GC was higher among resettlers than in the general population in Münster [24]. In the analysis of the Saarland and Münster cohort combined as well as separate, the incidence of intestinal GC among resettlers remained elevated in the second period of the observation time and no secular trend could be observed. Regarding the incidence in the general population in Münster, ASRs illustrated a decreasing trend of total GC and other/missing type of GC, whereas no significant trend for intestinal or diffuse GC could be seen.

Over time, risk behaviors of migrants are expected to adapt to the ones in the host population, resulting in converging rates of morbidity and mortality [5]. This was the case for several cancer types among resettlers in Germany according to Kaucher et al. [25]. Initially, the incidence of all malignant cancer types was lower among resettlers than the general German population. Over time, the risk for cancer types such as colorectal, breast, and prostate cancer increased over time, towards the risk in the general population. However, for the elevated incidence of gastric cancer, no converging trend could be seen, possibly due to the small sample size and low number of cases. These findings were consistent with previous research, where migrants from non-Western countries were less prone to develop colorectal, prostate, and breast cancer but more likely to develop infectious related cancer types such as gastric cancer [4,26]. The converging trend for the former cancer types among resettlers may be explained by the adaptation to Western lifestyle (poor diet, sedentary lifestyle, etc.) and/or improved screening/diagnostic possibilities [4,25,33].

Like other infectious-related cancer types, gastric cancer was found to be more prevalent in non-Western countries [4]. In 2018, the age-adjusted incidence was 29.4 for men and 8.8 for women in Russia, in contrast to 9.4 and 4.5 in Germany [34]. As discussed in the article of Jaehn et al. in 2016 [24], the strongest risk factor for GC, *H. pylori* infection, cannot alone explain the higher risk among resettlers for only intestinal GC given its equal association to both subtypes. The development of GC is described as a complex multifactorial process, and several studies claim that *H. pylori* plays a role in early transformation steps, causing chronic inflammation, but the transitions that follow are determined by environmental, bacterial, and host factors [20,35,36,37,38]. While host and bacterial factors are less likely to change due to migration, in this case from FSU to Germany, it is more likely to expect environmental factors to be influenced by lifestyle and dietary patterns in the new country of residence.

The persistent elevated risk for intestinal GC among resettlers indicates a remaining gap between the migrant group and the host population. The differential risk could be explained by dietary patterns and lifestyle factors, which overall seem to be more strongly correlated to intestinal than diffuse GC. Studies show that heavy alcohol consumption is a risk factor, while intake of a diet rich with fruit and vegetables is a protective factor more strongly correlated to intestinal than diffuse type [21,22]. According to comparative studies [39,40], resettlers were more likely to be obese and less likely to take part in cancer screening. Smoking habits differed by sex, with female resettlers smoking less than the German population whereas male smoking more. Furthermore, resettlers consumed smaller amounts of alcohol but stronger alcohol. Resettlers ate more meat and potatoes, but no difference in vegetable intake was seen. Based on these findings, a higher consumption of stronger alcohol could increase the risk for developing intestinal GC, whereas obesity could increase the risk of GC in general. Moreover, resettlers have been shown to have lower socioeconomic status (SES) in comparison to the native German population, but with a decreasing gap over time [41]. Lower SES is known have a negative impact on health status through pathways and mechanisms understood to a certain extent [42]. A less beneficial lifestyle and dietary patterns could have an adverse effect on the risk of developing intestinal GC. However, generalization can be misleading as resettlers are a relatively heterogenous group, originating from different states and regions and possibly bringing divergent lifestyle and dietary habits with them.

### 4.2. Shortcomings and Limitations

The study is based on data that do not provide information on dietary pattern, lifestyle factors, etc. Therefore, it does not allow analyzing the association between different risk behavior and cancer diagnosis among resettlers. As cancer registration in the Münster region was not sufficiently complete before 1994, we decided to limit the analysis to the calendar period 1994–2013. Regarding the classification of GC cases, there was no validation study available on how to apply the Laurén scheme on the ICD-O-3 morphology classification. We used the same criteria as the Saarland study which was based on previous literature [24,43,44] and believe that a possible misclassification of histologic types of GC among resettlers and the general population would be non-differential, leading to a bias towards the null of SIRs.

The decreasing trend of other/missing GC in the general Münster population is most likely explained by both increased completeness of data and improved diagnostics techniques to classify histologic patterns of GC. The cases with missing histologic type are likely one of the two main subtypes and considering the median age at diagnosis for this group (77 years in general population, 70.5 for resettlers), these cases are more likely to be of intestinal type which is more common in older ages. The number of missing GC is larger in the beginning than the end of the observation time. If assuming that most of these cases would be of intestinal type, there could be a decreasing time trend hidden.

Selection bias is unlikely as all ethnic Germans were invited to Germany and a majority of them immigrated back and were allocated quasi-randomly into federal states [45]. This fact also makes bias caused by healthy migrant effect unlikely. The proportions of resettlers with ethnic German background decreased from 78% in 1993 to 19% in 2004 due to larger immigration of non-ethnic German family members and relatives [46]. The altered proportions may have increased the heterogeneity regarding characteristics and risk factor patterns in the migrant group. The reference population comprises both resettlers included and excluded in the cohort who are resident in the district of Münster. Due to the contamination of the group, real differences are less likely to be observed as ratios will move toward zero.

Due to data protection concerns, neither information on date of immigration nor mortality among individuals of the Münster cohort was available, which prevented us from analyzing the incidence among resettlers with respect to lengths of stay in Germany. However, as the majority of resettlers migrated to Germany in the first half of the 1990s, calendar time is highly correlated with length of stay. It should also be mentioned that person-time of the Münster cohort had to be estimated due to an incomplete follow-up in method thoroughly explained elsewhere [29].

### 4.3. Implications for Future Research

In conclusion, resettlers remain at higher risk of developing intestinal GC than the general population in Germany. Lifestyle and dietary patterns may be likely to explain the discrepancy as intestinal GC is more strongly associated with these factors. Given the poor prognosis for GC, prevention and early detection might ease the individual’s health burden as well as the society’s economic burden. We believe that further investigation of risk behaviors is needed to understand the development of disease pattern among migrants.

## Figures and Tables

**Figure 1 ijerph-17-09215-f001:**
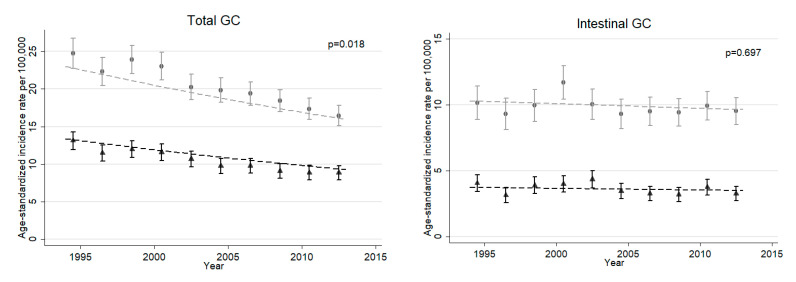
Age-standardized incidence rates (ASR) for each subtype of gastric cancer (GC) according to Laurén classification in the general Münster population. Plotted for two-year periods, from 1994–1995 to 2012–2013, with 95% confidence intervals. Males marked in gray, females in black. Dashed lines represent time trends modeled using negative binominal regression, with corresponding p-values of linear calendar year effect (rates are standardized with respect to the old European standard population).

**Table 1 ijerph-17-09215-t001:** Numbers of gastric cancer (GC) cases in the Münster population with corresponding ICD-O-3 ^a,b^ codes and classification by Laurén.

Laurén Classification	ICD-O-3 Codes	N ^c^
Intestinal GC (44%)	8140/3 Adenocarcinoma, not otherwise specified	2486
	8144/3 Adenocarcinoma, intestinal type	1774
	8211/3 Tubular adenocarcinoma	283
	8260/3 Papillary adenocarcinoma, not otherwise specified	66
	8480/3 Mucinous adenocarcinoma	169
Diffuse GC (26%)	8490/3 Signet ring cell carcinoma	1540
	8142/3 Linitis plastica	21
	8145/3 Carcinoma, diffuse type	1321
Other/Missing GC (30%)	Sections 802–857: Other carcinoma	697
	Sections 804 and 824: Endocrine carcinoma	182
	Sections 880–914: Sarcoma	159
	Sections 917–971: Lymphoma	1
	Section 800: Neoplasm	498
	Section 801: Carcinoma, not otherwise specified	1676
Total		10,873

^a^ International Classification System of Diseases for Oncology, 3rd edition; ^b^ Codes and terms can be unambiguously translated from ICD-O 2nd edition used 1992–2002; ^c^ N includes all gastric cancer cases among resettlers and the general population in Münster.

**Table 2 ijerph-17-09215-t002:** Characteristics of resettlers and the general population in Münster, and of resettlers and the general population in Münster and Saarland combined (pooled data).

	Münster (1994–2013)				Pooled Data (1990–2013)			
	Male [N (%)]		Female [N (%)]		Male [N (%)]		Female [N (%)]	
	Resettlers	Münster	Resettlers	Münster	Resettlers	Population	Resettlers	Population
Incident gastric cancer cases	51	6161	36	4712	87	8814	65	6914
Median age at diagnosis (Interquartile range)	66 (52–75)	70 (61–78)	72.5 (57–78)	76 (65–83)	68 (58–75)	70 (61–77)	71 (51–77)	76 (66–83)
Laurén classification								
Intestinal GC	28 (55)	3033 (49)	19 (53)	1745 (37)	53 (61)	4400 (50)	32 (49)	2634 (38)
Diffuse GC	14 (27)	1442 (23)	6 (17)	1440 (31)	17 (19)	2204 (25)	16 (25)	2198 (32)
Other/missing GC	9 (18)	1686 (28)	11 (30)	1527 (32)	17 (20)	2210 (25)	17 (26)	2082 (30)

GC, Gastric cancer; N, Number of cases.

**Table 3 ijerph-17-09215-t003:** Standardized incidence ratios (SIR) for subtypes of gastric cancer of resettlers compared to the general Münster population.

	Male		Females		Both Sexes	
	Observed	SIR (95% CI)	Observed	SIR (95% CI)	Observed	SIR (95% CI)
Total GC	51	1.50 (1.12–1.98)	36	1.32 (0.93–1.83)	87	1.42 (1.14–1.76)
Intestinal GC	28	1.64 (1.09–2.37)	19	1.91 (1.15–2.98)	47	1.73 (1.28–2.31)
Diffuse GC	14	1.61 (0.88–2.70)	6	0.64 (0.23–1.39)	20	1.11 (0.68–1.71)
Other/Missing GC	9	1.10 (0.50–2.09)	11	1.40 (0.70–2.50)	20	1.25 (0.76–1.92)

GC, gastric cancer; SIR, standardized incidence ratio; CI, confidence interval.

**Table 4 ijerph-17-09215-t004:** Standardized incidence ratios for subtypes of gastric cancer among resettlers in comparison to the general population in Münster and Saarland.

	1990–2001 ^a^		2002–2013 ^b^		Total	
	Observed	SIR (95% CI)	Observed	SIR (95% CI)	Observed	SIR (95% CI)
Total GC						
Male	31	1.87 (1.27–2.65)	56	1.72 (1.30–2.23)	87	1.77 (1.42–2.18)
Female	28	2.09 (1.39–3.03)	37	1.46 (1.03–2.01)	65	1.68 (1.29–2.14)
Total	59	1.97 (1.50–2.54)	93	1.60 (1.29–1.96)	152	1.73 (1.46–2.03)
Intestinal GC						
Male	19	2.45 (1.47–3.82)	34	1.98 (1.37–2.76)	53	2.12 (1.59–2.77)
Female	10	2.18 (1.05–4.01)	22	2.25 (1.41–3.41)	32	2.23 (1.53–3.15)
Total	29	2.35 (1.57–3.37)	56	2.08 (1.57–2.70)	85	2.16 (1.73–2.67)
Diffuse GC						
Male	5	1.15 (0.37–2.69)	12	1.31 (0.68–2.28)	17	1.26 (0.73–2.01)
Female	5	1.07 (0.35–2.49)	11	1.13 (0.57–2.03)	16	1.11 (0.64–1.81)
Total	10	1.11 (0.53–2.04)	23	1.22 (0.77–1.83)	33	1.18 (0.81–1.66)
Other/Missing GC						
Male	7	1.55 (0.62–3.20)	10	1.61 (0.77–2.96)	17	1.59 (0.92–2.54)
Female	13	3.17 (1.69–5.41)	4	0.67 (0.18–1.73)	17	1.69 (0.99–2.71)
Total	20	2.32 (1.42–3.59)	14	1.15 (0.63–1.93)	34	1.64 (1.13–2.29)

GC, gastric cancer; SIR, standardized incidence ratio; CI, confidence interval. ^a^ 256,089 person-years. ^b^ 411,101 person-years.

**Table 5 ijerph-17-09215-t005:** Negative binomial regression modeling standardized incidence rate ratios for gastric cancer among resettlers in comparison to the direct host populations in the Münster and Saarland cohorts.

Variable	Coefficient ^a^	95% CI	*p*-Value
Year (calendar year–1990)	0.002	−0.030, 0.035	0.889
Subtype			0.013
Intestinal GC	Ref.		
Diffuse GC	−0.639	−1.062, −0.215	
Other/Missing GC	−0.224	−0.648, 0.200	
Sex			0.886
Male	Ref.		
Female	0.025	−0.318, 0.368	
Cohort			0.003
Münster	Ref.		
Saarland	0.558	0.195, 0.921	
Constant	0.535	−0.053, 1.123	0.075

GC, gastric cancer; Ref, reference; CI, confidence interval. ^a^ The coefficients represent the change in the log of the SIR by one unit change of the calendar year—1990 or relative to the reference group.

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
