# Peer review of "The Incidence of Intestinal Gastric Cancer among Resettlers in Germany—Do Resettlers Remain at an Elevated Risk in Comparison to the General Population?"

_ijerph, 2020, doi:10.3390/ijerph17249215_

Round 1

Reviewer 1 Report

I do not feel the manuscript has been improved sufficiently. My previous concerns still remain unanswered. Manuscript in current form is not suitable for publication.

Reviewer 2 Report

The authors reviewed and made the necessary adjustments. I consider the article suitable for publication.

Reviewer 3 Report

Epidemiological research especially when it concerns cancers is tedious and difficult.  It is nonetheless very important and necessary to understand the biology and pathogenesis of cancers.  I commend the authors for their elaborate work and I do feel that the work is of interest to scientific community.

This study does have limitations and authors themselves had described the limitations in the article. 

‘A study on dietary patterns in the Russian population found that a salty diet was more prevalent in the Volga region, which was populated by native Germans suggesting resettlers also have a food culture rich in salt [41].’ It would be appropriate to remove this line.  Any relation to the article in question is tenuous as best.

Round 2

Reviewer 1 Report

Justification by authors to my comments is reasonable. I am sure their new study will study in detail about cancer progression in migrant population. Current study is suitable for publication in this journal.

This manuscript is a resubmission of an earlier submission. The following is a list of the peer review reports and author responses from that submission.

Round 1

Reviewer 1 Report

It is a robust work, which fulfills the objective and scope of the journal.
It is an innovative article on the incidence of GC in the immigrant population in Germany and possible related factors. However, the objectives are well defined and exposed in the results. And the limitations of the work, well addressed by the authors, delimit the field of study and exclude the possibility of assumptions without scientific basis. However, I ask the authors to make these small adjustments:
- Summary although concise, it should address the central theme of the research before the objectives are exposed.
- Keywords must be redone. Remember that the terms that are already in the title should not be repeated in the keywords.
- References are not in accordance with the standards recommended by the journal.

Reviewer 2 Report

The manuscript requires major revision. It is not clear and concise. The different sections should be revised and organized better. While the title of the manuscript seems interesting, the authors have not justified why this work was carried out and why it is important. Similar work has been done recently so why is this particular topic important and what novel information does this manuscript provide. Why is this population important to study? why the two cohorts?  There is no power calculation indicating the sample size. Perhaps the authors should consider the use of a checklist to report their findings.  The presentation of findings can be improved. It does not really provide demographic population statistics.The cancer is associated with lifestyle behaviours which are not presented anywhere in the manuscript until towards the end of the manuscript. 

Reviewer 3 Report

To,

Editor

Anna Lindblad and group have carried out extensive survey to assess risk of intestinal gastric cancer in resettler population in Germany. They analyzed incidences of GC occurrences in Munster cohorts and found elevated levels. Statistical analysis to conform elevated levels of GC cases in resettler population is confirmed by data provided. Data support the authors claim. However, authors do not provide any other data to support why GC incidences are higher as compared to host population. Although authors discuss extensively about role of dietary factors and lifestyles on GC occurrences but all of that is speculative. I feel that current study can be strengthened by provided some sort of reasoning behind elevated GC occurrences. I understand this approach is extensive but will definitely strengthen data and shed light on reason for occurrences of not only GC but other cancers. I will be happy to review manuscript after authors make necessary changes.

Reviewer 4 Report

In the present research, authors have investigated differences in intestinal gastric cancer among German resettlers in comparison to general population by replicating previous findings in a new cohort of resettlers and then by combining all available data. The methodology is good but lacks novelty. The AMIN and AMOR cohorts have been analyzed previously by researchers and authors have already referenced them in the text (including eight articles from two authors of this manuscript).

How the results compare to findings from another similar study, done on same two cohort for multiple cancers, by Simone et al (PMID: 30254988)?  

In the same study, author used logistic regression, instead of negative binomial. What was the rationale behind using NBR in this study?

How is the data distribution of cohort used in this study?

Would using another recommended method (eg. PMID: 30342488) change the outcome?